

# Nitrogen dioxide stratospheric column at the subtropical NDACC station of Izaňa from DOAS, FTIR and satellite instrumentation

Cristina Robles-Gonzalez[1,*], Mónica Navarro-Comas[1], Olga Puentedura[1], Matthias Schneider[3], Frank Hase [3], Omaira Garcia[2], Thomas Blumenstock[3] and Manuel Gil-Ojeda[1]

[1]Atmospheric Research and Instrumentation Branch. National Institute for Aerospace Technology (INTA), Ctra. Ajalvir s/n, Torrejón de Ardoz, 28850 Madrid, Spain.
[2] Izaña Atmospheric Research Center (IARC), Meteorological State Agency (AEMET), Tenerife, Spain
[3]Institute of Meteorology and Climate Research - Atmospheric Trace Gases and Remote Sensing. Karlsruhe Institute of Technology, Karlsruhe, Germany
[*]Current address: Meteorological State Agency (AEMET), Madrid, Spain

*Correspondence to*: C. Robles-Gonzalez (croblesg@aemet.es)

**Abstract.** A 13-years analysis (2000-2012) of the $NO_2$ vertical column densities (VCD) derived from ground-based (GB) instruments and satellites has been carried out over the Izaña NDACC (Network for the Detection of the Atmospheric Composition Change) subtropical site. Ground-based DOAS (Differential Optical Absorption Spectroscopy) and FTIR (Fourier Transform InfraRed spectroscopy) instruments are intercompared to test mutual consistency and then use for validation of stratospheric $NO_2$ from OMI (Ozone Monitoring Instrument) and SCIAMACHY (SCanning Imaging Absorption spectroMeter for Atmospheric CHartographY). The intercomparison has been carried out taking into account the various differences existing in instruments, namely temporal coincidence, collocation, sensitivity, field of view, etc. The paper highlights the importance of considering an "effective solar zenith angle" instead of the actual one when comparing direct sun instruments with zenith sky ones for a proper photochemical correction. Results show that $NO_2$ vertical column densities mean relative difference between FTIR and DOAS instruments is 2.8 ± 10.7 % for AM data. Both instruments properly reproduce the $NO_2$ seasonal and the interannual variation. Mean relative difference of the stratospheric $NO_2$ derived from OMI and DOAS is -0.2 ± 8.7 % and from OMI and FTIR is -1.6 ± 6.7. SCIAMACHY mean relative difference is of 3.7 ± 11.7 % and -5.7 ± 11.0 % for DOAS and FTIR, respectively. Note that the days used for the intercomparison are not the same for all the pairs of instruments since it depends on the availability of data. The discrepancies are found to be seasonally dependent with largest differences in winter and excellent agreement in the spring months (AMJ). A preliminary analysis of $NO_2$ trends has been carried out with the available data series. Results show positive values in all instruments but larger values on the ground-based than that expected by nitrous oxide oxidation. The possible reasons of the discrepancy between instruments and the positive trends are discussed in the text.

## 1 Introduction

Nitrogen dioxide ($NO_2$) is both a natural and anthropogenic constituent of the terrestrial atmosphere. In the stratosphere it plays an important role in the equilibrium of ozone through auto-catalytic cycles (Crutzen, 1970) and by deactivating other ozone depleting substances into their reservoir forms. In remote unpolluted regions, the most important contribution to the $NO_2$ vertical column comes from the stratosphere. The spatial and





vertical distribution in these regions was first studied in the 1970' by ground-based differential photometry and

spectroscopy (Brewer et al., 1973; Kerr and McElroy, 1976; Noxon, 1975). The measurements by differential

optical absorption spectrometry at zenith, or DOAS technique (Platt and Stutz, 2008) was since that time used

in remote locations for $NO_2$ long term monitoring. Using infrared spectral domain, Fourier Transform Infrared

Spectroscopy (FTIR) instrumentation was deployed in order to analyse atmospheric trace gases (Hendrick et

al., 2012). Few decades ago, both instrumental techniques joined together in the NDACC (Network for the

Detection of Atmospheric Composition Change) (http://www.ndsc.ncep.noaa.gov), a network developed to

provide accurate and standardized long-term measurements of atmospheric trace gases.

The recent needs of having reliable data in almost near real time to feed the forecasting models such as MACCII

(https://www.gmes-atmosphere.eu/) makes the comparison of observations performed by using different

techniques and the validation of satellite data using ground based observations a valuable tool to assure the

quality of both GB and satellite data and thus improve the model performance.

However, the intercomparison of remote-sensing instruments collected by independent instrumentation is not

straightforward. A deep knowledge of the representativeness of the scanned air masses provided by each

instrumental technique is required. In last years, a number of studies have focused on the comparison of satellite

$NO_2$ observations with GB instruments (FTIR and DOAS) in order to identify and quantify potential

discrepancies. Gil et al., 2008 published a climatology of the stratospheric $NO_2$ over the NDACC subtropical

station Izaña and a preliminary comparison with SCIAMACHY satellite instrument finding good agreement

between them (1.1% differences). Dirksen et al., 2011, compared more than 5 years (October 2004 to May

2010) of OMI stratospheric $NO_2$ from the OMI standard products (SP) and from the DOMINO algorithm with

NDACC remote stations GB measurements finding a mean difference of 13 %. Pinardi et al., 2011 reported an

agreement between GOME-2 and NDACC/UV-Vis network over the Northern Hemisphere within 8-20 %

depending on the season and latitude. Adams et al., 2012, presented an intercomparison of GB and satellite

$NO_2$ columns at a Polar Canadian station (PEARL). The satellite data they used were OSIRIS, ACE-FTS v2.2

and ACE-FTS v3.0 which agreed with GB measurements within 20 %. However, very few publications include

FTIR data in their comparisons.Wetzel et al., 2007, presented the validation of MIPAS-ENVISAT $NO_2$ data.

They showed that the mean deviation between the FTIR measurements and MIPAS from July 2002 until March

2004 remains within 10 % in Kiruna (68° N) and over Harestua (60º N) a mean negative bias of 15 % have

been presented for MIPAS- UV-vis daytime comparisons. Hendrick et al., 2012, compared stratospheric $NO_2$

datasets from the DOAS based instrument SAOZ and FTIR based instrument Bruker with satellite DOAS

instruments, namely GOME, SCIAMACHY and GOME-2 at northern mid-latitudes over Jungfraujoch from

1990 to 2009. They observed a FTIR minus DOAS GB mean relative differences of about -7.8 ± 8.2 % and

satellite minus DOAS GB mean relative differences of 0.9 ± 8.8 % for GOME, 1.9 ± 11.5 % for SCIAMACHY

and 2.3 ± 11.6 % for GOME-2. Recently, Belmonte-Rivas et al., 2014 have revised the stratospheric $NO_2$ data

retrieved from satellite instruments. They found discrepancies in stratospheric $NO_2$ obtained on nadir mode

when compared with the "limb" ones. Globally, SCIAMACHY was underestimated by 0.5E15 whereas OMI

data were found to be 0.6E15 too large. They also found a temperature dependence affecting the retrieval via

the AMF. Marchenko et al., 2015 and van Geffen et al., 2015, carried out a thorough revision of the *a-priori*

data used in the algorithms to retrieve $NO_2$ from OMI data. They concluded that the *a-priori* data used in the

algorithms should be improved. Temperature variability in the subtropical stratosphere is small as compared



to higher latitudes. The amplitude of the seasonal wave in the mid stratosphere is of 4K peak to peak (Gil et
al., 2008), thus minimizing the temperature dependence impact in the satellite retrieval.

The goal of this paper is to extend the previous GB to satellite intercomparisons to lower latitudes, including
DOAS and FTIR ground based techniques. Thirteen years of data, 2000 to 2012, from the Izaña NDACC
subtropical station have been used for this purpose. Once the agreement of GB instruments is proved, their
measurements are being used for the validation of OMI and SCIAMACHY satellite observations. In this work
a new correction method with a high impact on photochemical active species is also introduced and applied to
minimize the effect of the different scanned airmass when different kind of observations are used.

The use of two ground-based independent measurements techniques is also helpful for long term studies since
confidence is gained when searching trends, which are usually small compared with the seasonal cycle.
Additionally, extra information can be obtained on the heights where trends are observed if the vertical
sensitivity of the instruments is not identical, as it is the case with DOAS and FTIR. Previous studies on $NO_2$
trends are not conclusive.

Gruzdev and Elokhov, 2009 found a hemispherical dependence on the sign of the stratospheric $NO_2$ trend. A
positive trend was found over the middle latitudes of the Southern Hemisphere in good agreement with
expectations, whereas over the Northern Hemisphere it was negative in disagreement with the increase of
emissions of nitrous oxide ($N_2O$), which is a precursor of $NO_2$. Hendrick et al., 2012, also found negative trends
over Jungfraujoch for the period 1990-2009. Gil-Ojeda et al., 2015, found a hemispherical and latitude
dependence on the sign and magnitude of the trend based on 4 GB DOAS stations. Northern latitudes display
a positive trend whereas trends are negative in the southern latitudes. The picture is consistent with MIPAS
trends analysis (i.e. Eckert and von Clarmann, 2014). At present, this subject is under debate.

The work is organized as follows; Sections 2 and 3 present the DOAS and FTIR techniques, respectively, their
advantages and limitations as well as the basics of the algorithms used to extract $NO_2$ column abundances.
Section 4 gives a brief review of the satellite instruments SCIAMACHY and OMI and the algorithms whose
results have been used along this work. In section 5 the main characteristics of the Izaña Observatory are
highlighted. The procedure for the intercomparison is explained in section 6 and the results and discussion can
be found in section 7.

## 2 DOAS: Technique and Instrument

As previously mentioned, the DOAS technique (Platt and Stutz, 2008) has been extensively used in last decades
to measure stratospheric $NO_2$ since the pioneering works of Noxon, 1975 and Syed and Harrison, 1981. The
technique is based on the analysis of the absorption of sky radiation by the gas under consideration providing
that the magnitude of the absorption structure varies with the wavelength. For species of interest with the load
of mass located in the stratosphere, the spectrum of the sky is taken during the twilight to enhance the effect of
the stratospheric absorption and minimize the tropospheric contribution. The analysis is based on a linear fit of
the log-ratio of the sky background intensity spectrum with respect to a non absorbing spectrum.

INTA (National Institute for Aerospace Technology, http://www.inta.es/atmosfera/33/menu.aspx) have been
carrying out measurements of $NO_2$ at the Izaña Atmospheric Observatory (IZO), managed by the
Meteorological State Agency (Agencia Estatal de Meteorologia, AEMET, Spain, http://izana.aemet.es/), since
1993 and in the framework of NDACC since 1998. Since then, two DOAS instruments have been covering the



period of measurements at Izaña Observatory with a proper overlapping in order to consolidate the data time
series. During the period 1998 to 2010, the RASAS spectrometer was in operation. The instrument is based on

an EGG&1453A 1024 photodiode array detector controlled by an EGG 1461 on a Jarrel-Ash Monospec 18
spectrograph. Scattered light at zenith was collected by a baffled cylinder through a quartz fibre bundle. A
diffraction grating of 600 grooves/mm provided a spectral range of 340–600 nm for $NO_2$ and $O_3$ observations
with an average full width at half maximum (FWHM) resolution of 1.3 nm. The spectrograph and detector
were housed in a thermostatised hermetic container keeping the spectrograph at a constant temperature

maintaining the alignment of the spectra with time. A more detailed description of the instrument can be found
in Gil et al., 2008.

Since 2010 the instrument was replaced by a MAX-DOAS (Multi Axis Differential Optical Absorption
Spectroscopy) capability spectrometer (RASAS-II). The spectral range is 415–530 nm covering the largest
$NO_2$ spectral bands. It is based on a Shamrok SR-163i spectrograph and a 1024×255 pixels DU420A-BU Andor

Idus CCD camera. A detailed description of RASAS II instrument can be found at Puentedura et al., 2012 and
Gomez et al., 2014.

The analysis of the spectra was performed using software developed at INTA based on the standard DOAS
technique. A detailed explanation of the analysis routine can be found in Gil et al., 2008. DOAS settings for
$NO_2$ column retrieval follow NDACC UV/Vis Group recommendations (Hendrick et al., 2012; Van

Roozendael et al., 2012). A set of 6 cross sections has been included; $O_3$, $NO_2$, $H_2O$ and $O_4$. The Raman
scattering cross-section was generated by the Win-DOAS package (Fayt and Van Roozendael M., 2001) from
the Raman theory. Finally, the inverse of the reference spectrum was included as a pseudo cross-section to
account for stray light inside the spectrograph and detector residual dark current. The air mass factor (AMF)
used for the conversion to $NO_2$ vertical column is based on the harmonic climatology of stratospheric $NO_2$

profiles developed by Lambert et al, 1999. Further details are shown in Table 1.

**3 FTIR. Technique and Instrument**

Ground-based FTIR measurements are performed at Izaña Observatory since 1999 (Schneider et al., 2005)
under a collaborative effort between KIT (Karlsruhe Institute of Technology, http://www.imk-
asf.kit.edu/english/) and the Spanish Atmospheric Research Centre of AEMET. It is part of the NDACC. In

2005 a Bruker IFS 125 HR spectrometer (García et al., 2012), was installed in a container and equipped with
a solar tracker at Izaña Observatory. The solar tracker is controlled by a camera and Camtracker software (Gisi
et al., 2011). Solar transmission spectra are recorded in the spectral range of 2 to 13 micrometre using InSb
and MCT detector. In order to improve signal to noise ratio the NDACC optical filter set is used. The
instrumental line shape (ILS) is monitored on a regular basis using cell measurements and LINEFIT software

(Hase et al., 1999). Spectra are analysed using PROFFIT retrieval code (Hase et al., 2004). PROFFIT includes
a forward calculation model and an inversion tool to retrieve profiles and column amounts of trace gases from
atmospheric spectra. The NDACC harmonized retrieval scheme is applied. As spectroscopic data the HITRAN
2008 line parameters (Rothman et al., 2009) are used. Daily NCEP data is used for pressure and temperature.
Profiles and column amounts of trace gases like $H_2O$, HDO, $CH_4$, $N_2O$, CFC-11, CFC-12, $O_3$, $HNO_3$, $ClONO_2$,

HCl, HF, NO, and $NO_2$ can be derived from the infrared spectra. For $NO_2$ a spectral microwindow, providing



weak $NO_2$ lines superimposed to a strong broad band methane absorption, around 2914.5 cm-1 is fitted. The $NO_2$ total column retrieved using this algorithm is mainly sensitive to the stratospheric abundance.

**4 Satellite Instrumentation**

**4.1 SCIAMACHY**

SCIAMACHY was a satellite imaging spectrometer on board of ENVISAT platform in operation from March 2002 to April 2012. It measured backscattered, transmitted or reflected radiation from the Earth surface and atmosphere with a moderately high resolution (0.2 nm to 1.5 nm) in the wavelength range of 240 and 1700 nm for global remote sensing of trace gases, aerosols and clouds. It measured in nadir, limb and occultation modes

(Bovensmann et al., 1999; Burrows and Chance, 1991) with a swath of 960 km across track with a resolution of 30 x 60 $km^2$ in the nadir mode. Detailed information about operation characteristics of SCIAMACHY can be found at http://www.sciamachy.org/. The stratospheric $NO_2$ is retrieved by using the DOAS technique in the spectral region of 425-450 nm. The data used in this paper have been generated by the Institute of Environmental Physics (IUP) of the University of Bremen algorithm v2.0 (http://www.iup.uni-

bremen.de/doas/scia_no2_data_acve.htm) (Sussman et al., 2005) based on the work of Richter and Burrows, 2002, for GOME $NO_2$. SCIAMACHY stratospheric $NO_2$ values within 200 km around the station are included in the dataset. $NO_2$ cross-sections used in the analysis are those by Vandaele et al., 1998, which are also used for DOAS retrieval. SCIAMACHY data have not been corrected for cross-section temperature dependence. Assuming a dependence of 0.3 to 0.5%/K (Bucsela et al., 2013, Boersma et al. 2005), for the Izaña latitude,

the maximum error due to this effect is of 1.2% to 2%.

**4.2 OMI**

OMI is a hyperspectral imaging instrument (Levelt et al., 2006) on board AURA that measures the backscattered Earth radiation in the UV-VIS spectral range (from 264–504 nm) with a spectral resolution between 0.42 nm and 0.63 nm. It has a spatial resolution of 13 km along-track by 24 km cross-track for the

nadir pixels. The swath width is about 2600 km providing daily global coverage.

The OMI stratospheric $NO_2$ data have been computed with the NASA Standard Product OMNO2 algorithm (version 2) (Bucsela et al., 2013) applying the DOAS technique in the spectral range of 405 nm to 465 nm (Boersma et al., 2002; Bucsela et al., 2006). As in the case of SCIAMACHY, OMI stratospheric $NO_2$ values within 200 km around the station are included in the dataset. The cross-sections used in the analysis are $NO_2$

from Vandaele et al, 1998, $O_3$ are from Bass and Johnsten, 1975, and $H_2O$ from Harder and Brault, 1997. An empirical temperature correction factor is applied to the $NO_2$ absorption cross-sections similar to the factors used by Boersma et al. (2002, 2004). For more information about the stratospheric $NO_2$ scientific algorithm see Bucsela et al., 2013. In addition, a thorough revision of the spectral fitting algorithm is presented in Marchenko et al., 2015 and van Geffen et al., 2015. We used the data series available in the beginning of 2013.

From the results of Marchenko et al., 2015 and van Geffen et al., 2015, presumably the $NO_2$ values would be lower with a new algorithm.





**5 Izaña Atmospheric Observatory.**

The Izaña Atmospheric Observatory (IZO) is a high mountain NDACC station located on Tenerife island in the subtropical North Atlantic Ocean (28.3ºN, 16.5ºW, 2370 m a.s.l.), where DOAS and FTIR instruments are

in operation since 1993 and 1999, respectively. IZO is run by the Izaña Atmospheric Research Centre (IARC, www.izana.aemet.es), belonging to State Agency of Meteorology of Spain (Agencia Estatal de Meteorología, AEMET).

IZO is located above a quasi-permanent temperature inversion layer established between 800 and 1500 m a.s.l. associated to the trade-winds regime. The inversion layer separates the moist marine boundary layer from the

dry free troposphere and works as a natural barrier for local and regional pollution (Cuevas et al., 2015 and references therein). Thereby, the $NO_2$ VCD measured with GB instruments at IZO can be considered as stratospheric $NO_2$ without anthropogenic influence.

**6 Comparison Methodology**

The signal of remote sensing instruments using direct or diffuse solar radiation as a source is a weighted average

of rays crossing the entire atmosphere through different paths. The averaging kernels (AVK) matrix defines the relation between the retrieved quantities and the true atmospheric state (Eskes and Boersma, 2003; Rodgers, 2004) and it can be viewed as the sensitivity of the instrument to the trace gas in the different layers.

Large differences in averaging kernels profile between DOAS, FTIR, and satellite techniques would lead to uncertainties difficult to quantify and could result in a more complicated intercomparison (Adams et al., 2012;

Dirksen et al., 2011; Hendrick et al., 2012; Peters et al., 2012; Sussmann et al., 2005, among others). This, however, is not the case. Figure 1 shows how all considered instruments have their maximum sensitivity in the stratosphere, and how the tropospheric effect is minimum. AVK are plotted for the diurnal period of measurements of each instrument, DOAS at solar zenith angle (SZA) of 89º-91º, FTIR for SZA 50º am to 50º pm and satellites around noon. The DOAS tropospheric response is almost zero since at twilight the effective

scattering height, which is the height where the rays are scattered downward to the instrument, is located in the lower stratosphere. The contribution of the lower layers of the troposphere to the nadir satellites signal is also small since scattering is still large at 440 nm. In any case, satellite algorithms take into account the effect of the troposphere to eliminate it from the stratospheric $NO_2$ results to avoid potential pollution episodes. The $N_2O$ signatures used by the FTIR retrieval are rather weak. Therefore, the retrieval approach of a scaling a-

priori profile is used. Due to the fact that the absorption contribution of stratospheric $NO_2$ is less pressure-broadened, it has per molecule a larger impact on the least-squares fit than a tropospheric contribution. In addition, the $NO_2$ signatures are surrounded by strong $CH_4$ lines, and the imperfect spectroscopic description of the wings generated by these lines requires the fit of additional empirical background parameters, which results in a further decrease of the retrieval sensitivity with respect to tropospheric $NO_2$. Therefore all

intercompared instruments are highly sensitive at the altitude range where the $NO_2$ bulk is located, and minimize the potential differences near the surface that may occur due to pollution events. From the AVK assessment it can be seen that $NO_2$ columns measured by nadir satellites, DOAS and FTIR can be directly compared. Being the station in a remote location in the free troposphere at 2370 m a.s.l., well above the major



source of pollution, the column data are representative of the stratospheric $NO_2$. For that reason, the satellite

product "stratospheric column" has been used here.

The algorithms to generate atmospheric products from DOAS and FTIR instruments require an "a priori" $NO_2$ profile. DOAS employs the $NO_2$ climatology obtained from a Fourier Harmonic decomposition of UARS HALOEv19 and SPOT-4 POAM-IIIv2 profiles data (Lambert et al., 1999) whereas FTIR utilizes the output of the WACCM climatic model. To test the influence of the profile used on the final products, DOAS AMF have

been obtained by means of the WACCM profile for a case study. Results of the comparison show that the selection of the selection of the profiles has a maximum impact of 6 % on the retrieved columns.

The importance of a proper collocation when intercomparing instruments from different platforms and techniques has recently been recognized. The spatial coincidence, the field of view, the data vertical and horizontal smoothing as well as the location of the effective airmass has to be taken into account. The

instruments should observe the same airmass in the atmosphere (spatial coincidence). This requirement is not always easy to fulfil when comparing instruments based on different techniques. In cases where the spatial coincidence is not possible, the comparison of the instruments can be performed taking into account the impact of such differences on the data. The "effective location" is the surface projection of the studied air parcel. For instance, DOAS instruments measure scattered light at zenith during twilight. As the stratospheric $NO_2$ layer

is located at about 25-30 km height (orange layer in figure 2a), the effective stratospheric $NO_2$ mass measured by the instrument is about 300 km toward the sun direction. Two effects are observed from this figure: the first one is that when satellite nadir data are to be compared with DOAS, the selection of the ground pixel for collocation must be done taking into account the DOAS effective airmass. Figure 2b shows the surface projection of the central point of the effective airmasses for DOAS and FTIR. DOAS zenith sky scans 300 km

towards the East and West in morning and dusk, respectively, whereas FTIR direct Sun measurements are representative of the stratosphere in the surroundings of the station. There is also dependence on the season: about 300 km in latitude change from summer to winter solstices. The size of the airmasses scanned by each instrument is also a factor affecting the degree of agreement. While FTIR scans a narrow field of view, satellites footprint cover an area of few decades to hundreds of squared km and for comparison all measurements falling

within a given area are considered. In our study a circle of about 300 km around IZO has been considered.

At the tropical-subtropical latitudes the $NO_2$ spatial variability in the stratosphere is low and the impact of these effects on the intercomparisons is small. From tests carried out over the area with SCIAMACHY data it has been found almost no zonal gradients and meridional gradients of $2.0 \times 10^{13}$ molec.cm-2 per degree, in subtropical unpolluted areas, representing 0.84 % per latitude degree of the column. Therefore the impact of

the changing scanned volume with season might have a maximum effect of 3 % of the column. Around the Canary Archipelago there are small longitudinal differences associated to coastal pollution that will be discussed in section 7.

The second effect is that the effective solar zenith angle at the effective airmass area is about 3º lower than the actual SZA at the station (local SZA). Figure 2c shows the local and the effective SZA of the observed

stratospheric $NO_2$ layer (in grey colour). In photochemically active species such as $NO_2$ a SZA dependent correction has to be applied to correct for the diurnal variation in the column and refer the measurements to a common time. This effect was already mentioned by Gil et al., 2008. In addition, Adams et al., 2012 observed that over polar areas the SZA dependent correction is particularly significant in spring and fall. As previously



mentioned, the effective DOAS $NO_2$ airmass is located in sun direction and the effective SZA (ESZA) at the

working latitude is 86.8º for AM and PM measurements. To compute it, the height of the centre of the
stratospheric $NO_2$ layer has been used.

In figure 3 the theoretical $NO_2$ photochemical behaviour over Izaña is shown together with the time overpasses
of the OMI and SCIAMACHY instruments. As the $NO_2$ VCD increases over the sunlight part of the day,
densities measured at different times must be corrected. In our work the stratospheric $NO_2$ from FTIR (AM

data), SCIAMACHY and OMI instruments has been photochemically corrected to the DOAS AM
measurement time using the BIRA-IASB (Belgian Institute for Space Aeronomy) stacked box photochemical
model PSCBOX (Hendrick et al., 2004, 2012), daily initialized with SLIMCAT 3D CTM chemical and
meteorological fields (Chipperfield, 2006) extracted for (30°N, 0°E) for the 2000-2009 period. Based on these
simulations, a climatology of $NO_2$ VCD diurnal variation is built and correction factors appropriate for the

photochemical matching between FTIR and DOAS observations are extracted from it. The ESZA angles have
been used for this purpose.

Another potential source of discrepancy might arise from the local meteorology diurnal variation. FTIR takes
measurements in the hours around noon, when the mountain anabatic winds transport airmasses from the
Marine Boundary Layer to the level of the station, as can be seen from "in-situ" measurements (Gil-Ojeda et

al., 2015). However, since the upwelling takes place in a shallow layer near the surface, the impact on the
overall column seems to be very low. DOAS instrument measures during twilight, when breeze if exist, is
small. In summary, we expect the diurnal upwelling to have a negligible impact in the comparison and have
not been considered here.

## 7 Results

Figure 4 displays the cross-correlation of FTIR, SCIAMACHY and OMI data versus DOAS considering the
SZA over the station and the ESZA for the collocation criteria correction described in previous sections. The
figures show that in all three cases the agreement strongly improves when the photochemical correction due to
temporal collocation is referred to the ESZA where the bulk of the $NO_2$ is located (86.8º) rather than the SZA
of the DOAS instrument (90º). In all cases ESZA corrected points fit better the diagonal for the range of $NO_2$

values. The slope is slightly reduced in all cases but the ESZA correction improves the comparison. In Figure
4, lower panel, the frequency analysis of the distance from the point P(X, DOAS) to the diagonal is represented,
being X all other instruments. In table 2 the parameters of the Gaussians and the linear fits are shown. The
mean distance with respect to the diagonal of the FTIR-DOAS represented by the Gaussian frequency
distribution improves from $-3.66 \times 10^{14}$ to $-3.77 \times 10^{13}$ molec.cm$^{-2}$, the OMI-DOAS from $6.02 \times 10^{14}$ to $-5.31 \times 10^{12}$

and the SCIA-DOAS from $2.23 \times 10^{14}$ to $-7.21 \times 10^{13}$ providing evidence that the photochemical correction has
to be applied for the ESZA, that is the SZA where the effective ray crosses the bulk of the $NO_2$ layer rather
than the local SZA at the observation point.

From now on, stratospheric $NO_2$ converted to 90º is not used any more in this work. All following results have
been computed with the ESZA correction.

The FTIR $NO_2$ photochemically corrected data are averaged to have one value per twilight. Figure 5a presents
the monthly mean variation of the $NO_2$ measured by the DOAS and the FTIR instruments. The lower values
on the plot correspond to AM values whereas the higher ones are PM $NO_2$ values. In Figure 5b the relative



difference between both instruments is presented in %. In general, results show a good agreement. AM values compare better than PM values with 2.8 ± 10.7 % and 11.7 ± 9.5 % respectively, even though mean values are

within the standard deviation in both cases. A fraction of the differences found in the PM data might be genuine since at noon and during the afternoon high $NO_2$ boundary layer airmasses are upwelled by the forced heating of the surface (upslope breeze) and can be seen by the direct Sun FTIR, even though its response is small to tropospheric pollution. Additionally, airmasses located to the West, in mid Atlantic, are representative of the background condition, whereas the East ones have slightly more $NO_2$ in the troposphere, contributing to 3-4

% larger columns (Figure 6). The treatment of the $NO_2$ diurnal variation in the box model used for the correction might have a contribution, as well, specifically the $N_2O_5$ photodissotiation rate. These results show the limitations existent when comparing remote sensing data obtained with independent techniques sampling non-identical air masses at non-identical times. Figure 5b also shows a change in the FTIR-DOAS behaviour at the beginning of 2005 both in AM and PM data. On that year the FTIR instrument was switched from a

Bruker IFS 120M to a Bruker IFS 120/5HR, less noisy than its predecessor. The improvement is observed by a decrease in the relative differences between instruments. It is also observed that FTIR $NO_2$ values are, in general, higher than DOAS $NO_2$ values. Dirksen et al., 2011 found similar results over Izaña. In contrast to Hendrick et al., 2012 that found FTIR measurements lower than DOAS SAOZ instrument by 7.8 ± 8.2 % on average over the NDACC Jungfraujoch station. Such differences are attributed to uncertainties related to the

respective spectroscopic parameters and differences in sensitivity profiles. Adams et al., 2012 compared FTIR results with SAOZ and PEARL (Polar Environment Atmospheric Research Laboratory) ground-based instruments operating in the UV and the VIS. They found that FTIR measures less $NO_2$ than the DOAS instruments by 12.2 ± 19.2 %.

Figure 7 shows the OMI comparison with DOAS and FTIR measurements. OMI stratospheric $NO_2$ data located

300 km around the Izaña station were used. Figure 7b shows the relative differences in % of satellite minus GB $NO_2$ values. In general, the results compare extremely well within -0.2 ± 8.7 % for the OMI-DOAS validation and -1.6 ± 6.9 % for the OMI-FTIR (see Table 3). Dirksen et al., 2011 validated OMI $NO_2$ with independent GB measurements from October 2004 to May 2010. Over Izaña, they presented OMI versus FTIR relative differences from 4 to 7 % which are similar to the present paper results while for the OMI-DOAS

relative differences are larger, in the range of 26 to 29 %. A possible reason for that difference is the photochemical correction applied to the $NO_2$ DOAS data to be compared with the OMI data. FTIR measurements used in this study, on the other hand, were close in time to the OMI overpass and no corrections to the data were made. Belmonte-Rivas et al., 2014 found that OMI derived stratospheric $NO_2$ values were higher than those obtained with the SCIAMACHY and GOME-2 instruments. They claim that the bias of the

OMI stratospheric $NO_2$ are due to the *a-priory* information used in the retrieval such as the absorption cross sections, spectral fit window width, spectral resolution, solar reference spectra, ring spectra, etc. Therefore, a thorough revision of the retrieval methodologies is necessary. In that sense, recently, major efforts have been done in order to improve the algorithms that compute the $NO_2$ from OMI data (Marchenko et al., 2015 and van Geffen et al., 2015). Marchenko et al., 2015 showing a reduction of the stratospheric $NO_2$ VCD of 20-30 %.

Van Geffen et al., 2015 found a reduction on the RMS of about 32%. Remarkably, in this work such discrepancies were not found. This issue is still open and further work is required to understand the discrepancies.





Even though the results agree well with the literature, a test exercise has been carried out to improve the quality of the validation exercise. Following the recommendation made in Section 6 about the effective air mass

(EAM) for the OMI/DOAS validation, instead of using all the OMI data retrieved 200 km around the station, we have only selected the data whose centre longitude fall in between the station and the sun (AM values). In that way the OMI measurements closer to the effective DOAS air mass are included in the validation. The new result of the validation is 0.2±8.6 % that presents a little improvement of the validation, (see in Table 3) proving that this effect is not crucial for the stratospheric $NO_2$ because its longitudinal variation is small. Therefore this

test exercise has not been applied to SCIAMACHY data that is presented next.

Stratospheric $NO_2$ dataset from SCIAMACHY has been compared with DOAS and FTIR data as well. The monthly mean interannual variation is presented in Figure 8a. The relative mean difference in % of the stratospheric $NO_2$ from SCIAMACHY, DOAS and FTIR instruments is shown in Figure 8b. Results show that SCIAMACHY agrees within -3.7 ± 11.7 % with DOAS and within -5.7 ± 11.0 % for the comparison with

FTIR results. A summary of the results are shown in Table 3. Note that the days used for the intercomparison are not the same for all the pairs of instruments since it depends on the availability of data. Results are reasonable, even though SCIAMACHY generates lower values than the ground-based instruments, in contrast with previous studies. Gil et al., 2008 reported SCIAMACHY minus DOAS differences of 1.1 % over Izaña, but being SCIAMACHY higher than DOAS values. Hendrick et al., 2012 also reports higher SCIAMACHY

$NO_2$ values than SAOZ over Jungfraujoch of 1.9 ± 11.5 %. The most probable reason for the discrepancies is related to the photochemical correction carried out to the SCIAMACHY results. In the present study, the ESZA has been used for the correction, leading to lower values of stratospheric $NO_2$ than those obtained using the local SZA in previous analysis. Taking into account the differences in techniques and time of measurements, SCIAMACHY and ground based data are in good agreement.

The agreement of the different instruments is found to be seasonally dependent (Figure 9). There is little scattering and differences around zero on spring months (AMJ), whereas discrepancy increases towards the winter months. All instruments remain within 10 % or better. Surprisingly, GB instruments behave differently than the satellite ones. OMI and SCIAMACHY show the seasonal maximum in June, quite in phase with the solar radiation. Ground-based instruments, on the contrary, display the maximum in July and large columns

are found in September as well. Gil et al., 2008, found the $NO_2$ column over Izaña for the period 1994-2005 to be modulated by the middle stratosphere temperature. However, the maximum of the secondary $NO_2$ cycle is located in March, and cannot explain the July peak. This effect is currently under study.

A preliminary estimation of stratospheric $NO_2$ VCD trends has been made by means of a linear regression calculation. In order to avoid the $NO_2$ seasonal dependence, the calculation has been performed for each month

(see figure 10). In the study the possible influence of the solar cycle and QBO are not included neither that of the stratospheric temperature. All instruments show positive trends in $NO_2$ stratospheric column. The DOAS instrument present the largest one with 10 to 15 %/decade significant at the 90% confidence level depending on the month for the period 2000 to 2012, while FTIR, OMI and SCIAMACHY trends show values up to 10 %/decade on individual months, but lower on average and with small confidence levels. The preliminary results

of the FTIR trend is significant at the 60 % of confidence level, OMI trend at the 80 % and SCIAMACHY at the 60 % of confidence level. On the annual mean, DOAS trend is larger by a factor of 3 than the rest of the instruments. These positive trends exceed by far those expected by the nitrous oxide oxidation. $N_2O$ is



increasing at a nearly steady rate of 2.5 %/decade. It is also of opposite sign that those found over the Jungfraujoch station, in the Swiss Alps (Hendrick et al., 2012). The observed trend could be slightly biased if

a trend in the stratospheric temperature has occurred during the analysed period. Recent report based on satellite most extensive data to date, found little or no trend in global lower stratospheric temperature since 1995 to 2013 (Seidel et al., 2015) whereas NCEP/NCAR reanalysis shows a decrease of 1.5K/decade at the 10 hPa level for the Tenerife area and for the 2000-2012 period (private communication). If confirmed, this effect could account for some 3% of the observed trend.

The observed increase at tropical-subtropical latitudes is in agreement with the analysis of the MIPAS global $NO_y$ data for the period 2002 to 2012 and the output of the WACCM model for the same period of time (Funke et al., 2015). Since the global $NO_y$ remains almost constant, the observed increase has been attributed to a displacement of the subtropical barrier as a consequence of stratospheric temperature changes (Eckert and von Clarmann, 2014). The maximum trend occurs in late winter and beginning of spring, supporting the dynamic

explanation. Figure 11 shows the ratio of sensitivities in the stratosphere between DOAS and FTIR, weighted by the concentration at each height. It is found that DOAS sensitivity is higher than FTIR in the lower stratosphere (below 28 km) whereas the opposite is true above that region. Larger trends in DOAS mean that the increase takes place in the layer in which $NO_x$ is dynamically controlled, also playing in favour of the above mentioned explanation. A more detailed study on $NO_2$ trends is ongoing in order to improve the preliminary

trends presented here and to better understand the results.

**8 Conclusions**

$NO_2$ total columns derived from two ground-based independent techniques, DOAS and FTIR from the NDACC network over Izaña (28º N, 16º W, 2370 m a.s.l.) have been intercompared for the period 2002-2012. Once mutual consistency has been proven, GB data have been used for OMI and SCIAMACHY validation of $NO_2$

stratospheric product. The paper discussed the concepts of spatial representativeness of the data and potential discrepancies related to differences in sampled airmass volumes, and time of measurements. The importance of the use of the effective solar zenith angle (ESZA) when comparing noon measurements with twilight measurements of photochemicaly active species is highlighted. For a gas with the bulk at 30 km height, the ESZA of a zenith DOAS with local SZA of 90º is 86.8º. After the correction of ESZA, the agreement between

instruments improves significantly, strongly reducing mean differences in all cases. The FTIR-DOAS mean difference of the data sets ranges from +2.8 ± 10.7 % to +11.7 ± 9.5 % for AM and PM data, respectively. Part of this difference is attributed to the photochemical box model used to reference to a common time of the day. For the satellite validation mean differences of -0.2 ± 8.7 % are found for the OMI/DOAS and -1.6 ± 6.9 for the OMI/FTIR comparisons. SCIAMACHY/DOAS shows a mean difference of about -3.7 ± 11.7 % and -5.7

± 11.0 for SCIAMACHY/FTIR. The seasonal cycle is well reproduced by all the instruments, with a dispersion increment during the winter months.

The agreement of the different instruments is found to be seasonally dependent. The differences are largest in winter months and almost disappear in spring (AMJ). Surprisingly, ground based instruments display the seasonal maximum in July whereas satellites show it in June. A preliminary linear correlation analysis shows

positive trends for all instruments above the rate of nitrous oxide oxidation. FTIR, SCIAMACHY and OMI mean annual trend is about 4 %/decade whereas DOAS observe 13.5 %/decade. This large discrepancy is



attributed to the DOAS high sensitivity to the lower stratosphere where the increase of dynamical origin seems to have been taking place.

Acknowledgements. This work has been carried out in the frame of the NORS (Demonstration Network Of ground-based Remote Sensing Observations in support of the Copernicus Atmospheric Service) project (funded by the European Community's Seventh Framework Programme (FP7/2007-2013) under grant agreement nº 284421; http://nors.aeronomie.be/) and has been partially supported by AMISOC project funded by the Spanish national funding Agency (CGL2011-24891). Authors acknowledge Dr. F. Hendrick from the

BIRA-IASB Institute for making available the photochemical correction used in this paper, the Institute of Environmental Physics of the University of Bremen for the SCIAMACHY $NO_2$ data and the Goddard Space Flight Center from NASA for the OMI NO2 data. Comments from the editor, Dr. Folkert Boersma, significantly improved the paper. We also acknowledge Emilio Cuevas for the temperature data series he sent us.

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





Tables.

**Table 1**. DOAS spectrometers settings

| Fitting interval | 450-533 nm |
|---|---|
| $NO_2$ cross-section | Vandaele et al. (1998), 220°K |
| $O_3$ cross-section | Bogumil et al. (2001), 223°K |
| $H_2O$ cross-section | Hitran (Rothman et al., 2008) |
| $O_4$ cross-section | RASAS spect. Greenblatt (1990), room temp.<br>RASAS-II spect. Hermans (1999), room temp. |
| Ring effect | Chance and Spurr (1997) |
| Orthogonalisation Polynomial | 3rd degree |
| Offset correction | Inverse of the reference |
| AMF calculation | BIRA-IASB $NO_2$ AMF LUTs |
| Determination of residual amount in reference spectrum | Modified Langley plot (Vaughan et al., 1997) |
| SZA range for twilight averaging of vertical columns | 89-91°SZA (Approx. 6 measurements) |


**Table 2.** Results of the linear fit of DOAS versus FTIR, OMI and SCIAMACHY $NO_2$ VCD for SZA=90º and ESZA=86.8º


|  | DOAS - FTIR | | DOAS -SCIAMACHY | | DOAS -OMI | |
|---|---|---|---|---|---|---|
|  | 90º | 86.8º | 90º | 86.8º | 90º | 86.8º |
| **LINEAR FIT** | | | | | | |
| Interc. | 1.12E15 | 0.92E15 | 0.77E15 | 0.64E15 | 1.41E15 | 0.82E15 |
| Slope | 0.77 | 0.66 | 0.81 | 0.70 | 0.74 | 0.66 |
| Adj. R-Square | 0.65 | 0.64 | 0.56 | 0.56 | 0.73 | 0.75 |
| **GAUSSIAN FIT** | | | | | | |
| Center | -3.66E14 | -3.77E13 | 2.23E14 | -7.21E13 | 6.02E14 | -5.31E12 |
| Width | 3.22E14 | 3.16E14 | 4.04E14 | 3.40E14 | 2.83E14 | 2.66E14 |

**Table 3.** Statistics of the relative difference of the stratospheric $NO_2$ from ground-based and satellite instruments in %.

|  | Nº | Mean | Standard deviation |
|---|---|---|---|
| **FTIR – DOAS (AM)** | 746 | 2.8 | 10.7 |
| **FTIR – DOAS (PM)** | 698 | 11.7 | 9.5 |
| **OMI – DOAS** | 2355 | -0.2 | 8.7 |
| **OMI – DOAS (EAM text)*** | 1298 | 0.2 | 8.6 |
| **SCIA – DOAS** | 1326 | -3.7 | 11.7 |
| **OMI – FTIR** | 540 | -1.6 | 6.9 |
| **SCIA - FTIR** | 314 | -5.7 | 11.0 |

[*] EAM = Effective Air Mass





Figures:

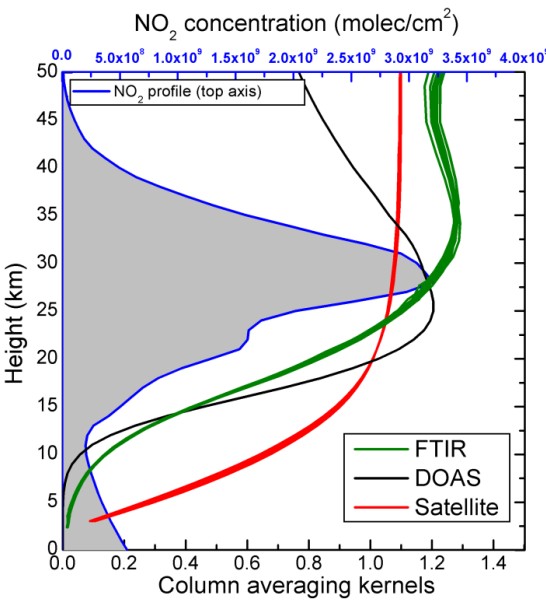

**Figure 1**.Typical averaging Kernels of DOAS (Black line), FTIR (green line) and satellite (red line)
instruments for the diurnal period of measurements. The shaded area represents a $NO_2$ vertical profile (top
axis) over Izaña obtained from the American Standard Atmosphere (U.S. Standard Atmosphere, 1976, U.S.
Government Printing Office, Washington, D.C., 1976) for the tropics






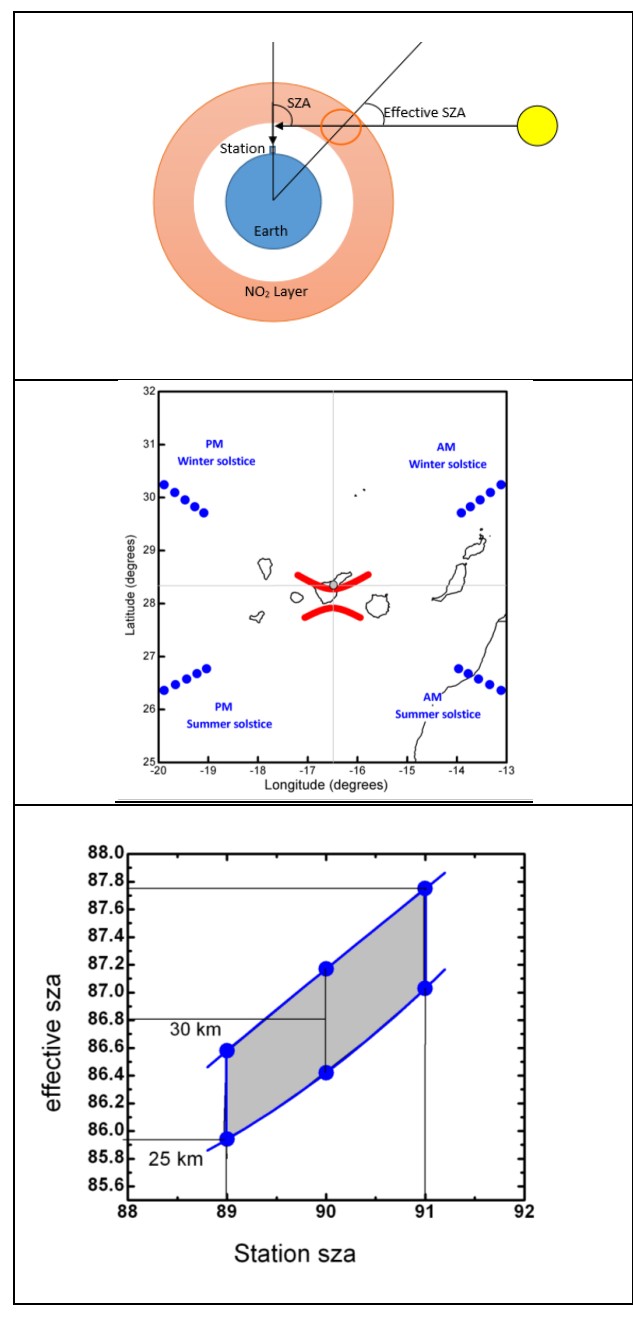

**Figure 2.** Representation of the effective NO$_2$ air mass (a) scanned by the DOAS instrument. (b) Surface projection of scanned airmasses for FTIR (red) and DOAS (blue) instruments for winter and summer solstices. Calculation assumes the bulk of the NO$_2$ layer at 30 km height. (c) Effective solar zenith angles versus local zenith angle for 25 km and 30 km NO$_2$ bulk height.






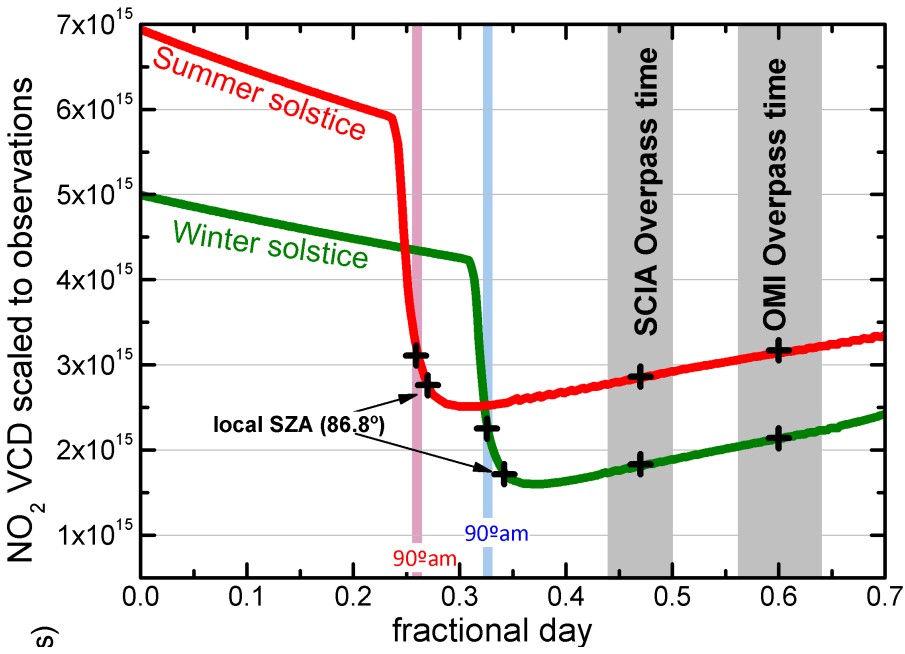


**Figure 3.** Example of the NO$_2$ photochemical behaviour from the SLIMCAT Box Model over Izaña. Red and green lines are the NO$_2$ VCD daily variation at the summer and winter solstices respectively. Grey areas are the time overpasses of SCIAMACHY and OMI instruments (Adapted from Gil et al., 2008)






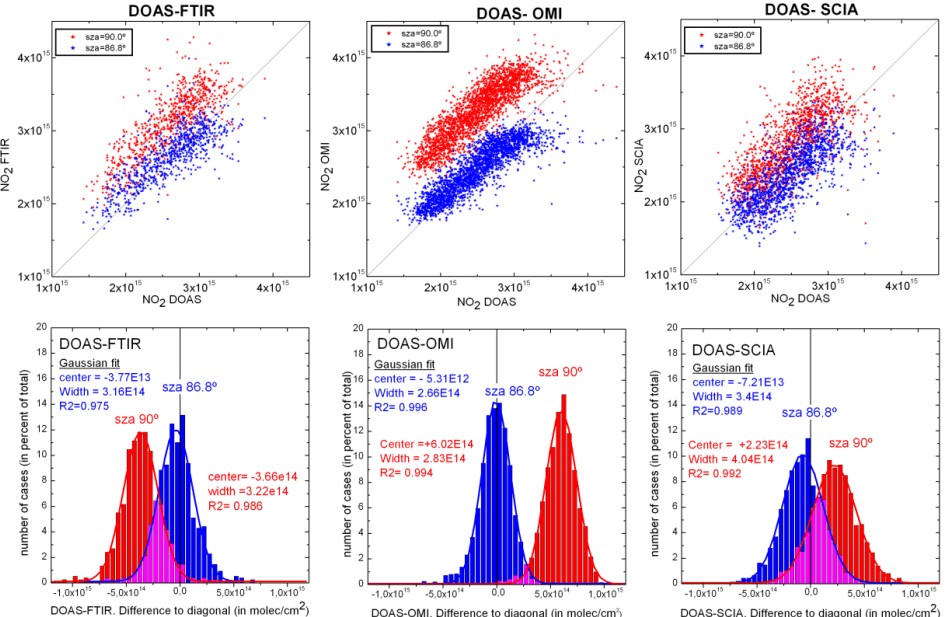

**Figure 4.** Upper panel. Scatter Plot of the NO$_2$ measured by DOAS instrument (X axis) and (a) FTIR, (b) OMI and (c) SCIAMACHY in the Y axis. The effective SZA = 90° is presented in red stars and effective SZA = 86.8° in blue stars. Solid diagonal represents the ideally perfect agreement. Lower panel. Frequency distribution of the distance of each cross-correlation point to the diagonal considering for the photochemical correction the local sza at 90° (red) and the effective sza at 86.8° (blue). See text for details.






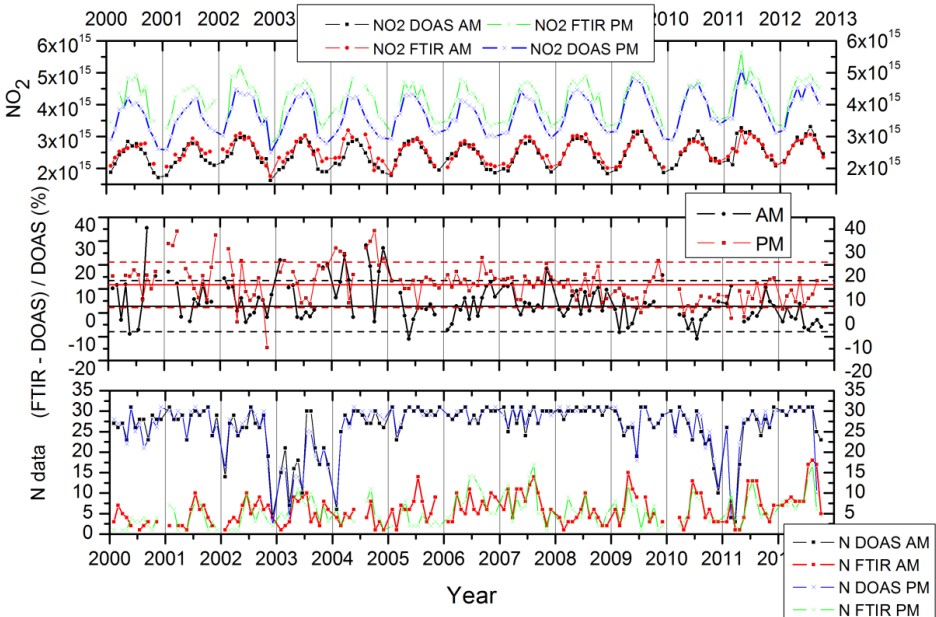

**Figure 5.** (a) Monthly mean evolution of NO₂ VCD from DOAS (black dots for AM, blue for PM) and FTIR (red dots for AM and green for PM). (b) Relative differences, solid line presents the mean relative difference and the dash lines represent the mean relative difference plus and minus the standard deviation in black for AM and in red for PM values. (c) Shows the number of days used to compute the monthly mean.






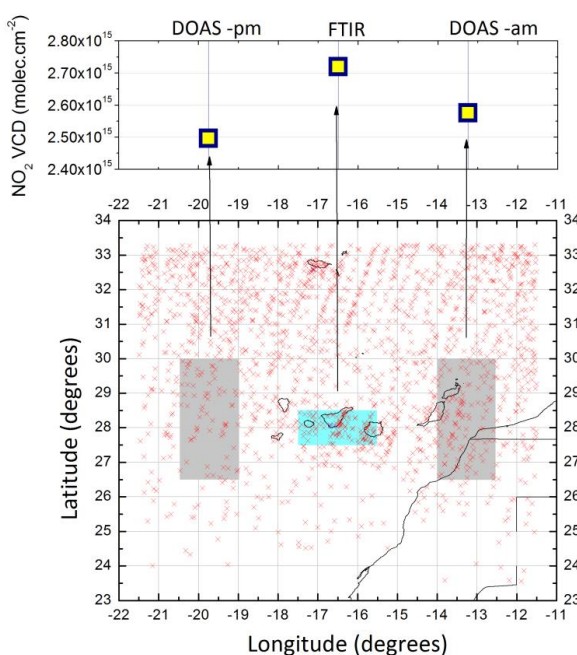

**Figure 6.** SCIAMACHY NO₂ vertical column density (VCD) annual mean over the areas where DOAS (grey rectangles) and FTIR (blue rectangle) are scanning the atmosphere.




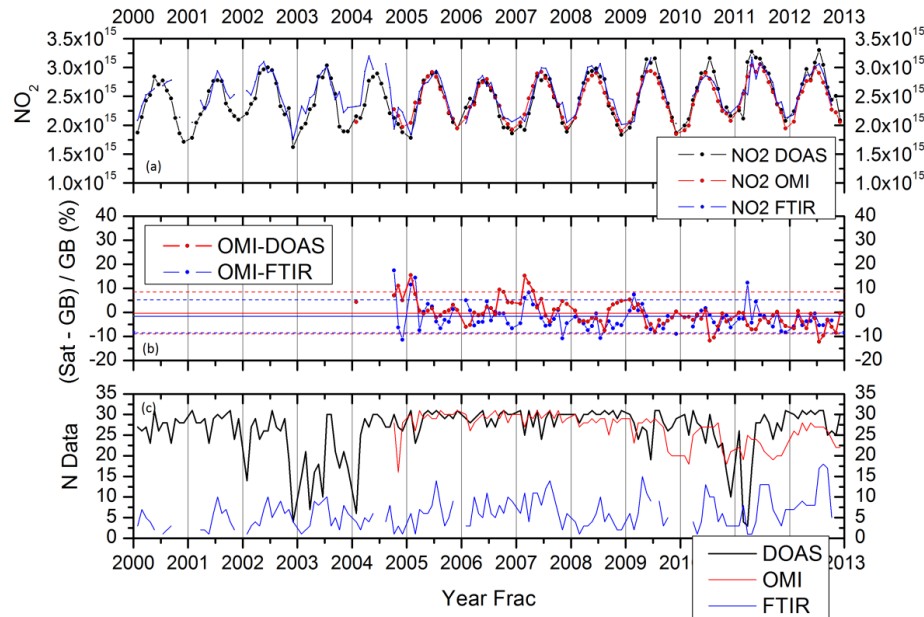

**Figure 7.** Same as Figure 5 but for OMI versus DOAS and FTIR (only AM data). Note that the scale in the top plot is different from Figure 5.

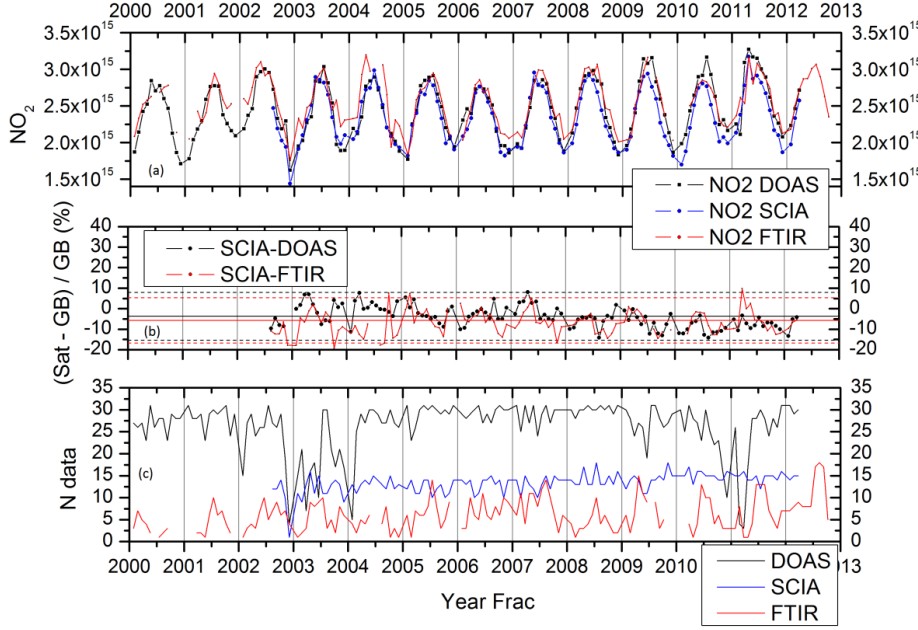

**Figure 8.** Same as Figure 5 but for SCIAMACHY versus DOAS and FTIR (only AM data). Note that the scale in the top plot is different from Figure 5.



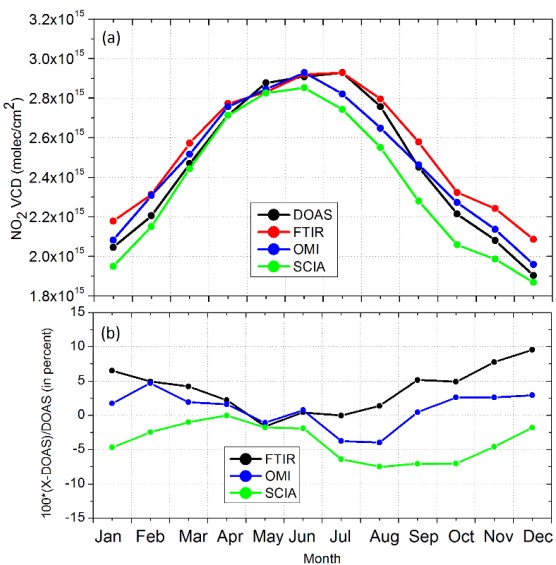

**Figure 9**. (a) Seasonal variation of $NO_2$ derived from DOAS (Black), FTIR (red), OMI (blue) and SCIAMACHY (blue-green). (b) $NO_2$ relative difference from the instruments with respect to DOAS.

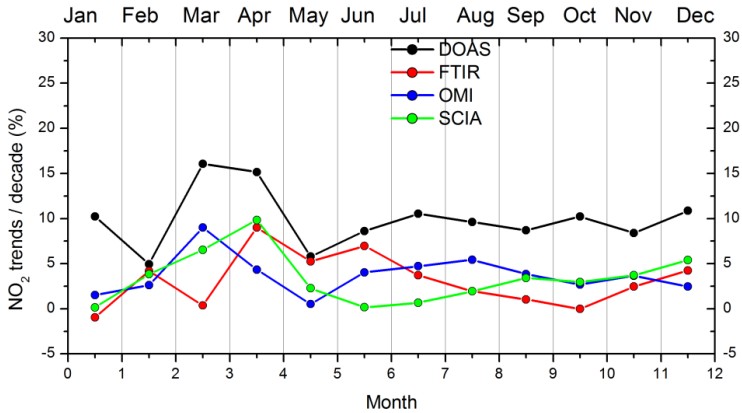

**Figure 10.** (a) $NO_2$ monthly trends from year 2000 to 2012. (b) Mean annual trends.





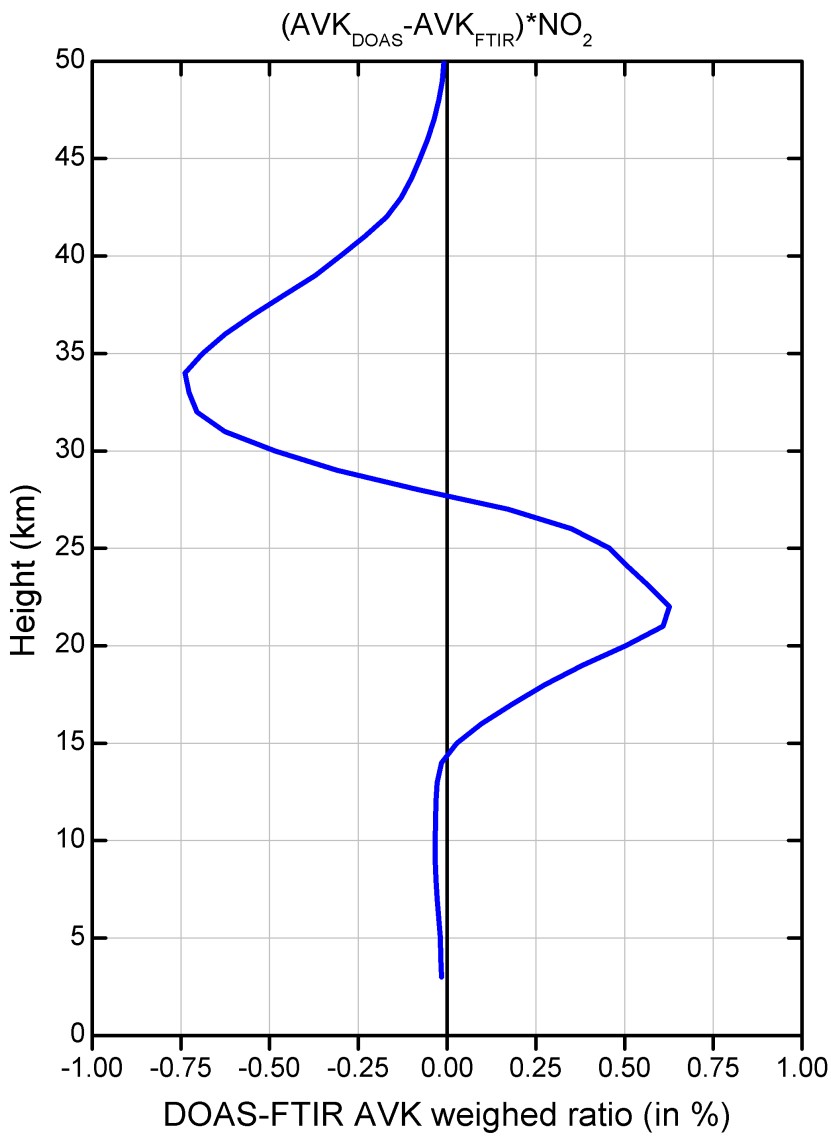

**Figure 11.** Ratio of sensitivities in the stratosphere between DOAS and FTIR weighted by the $NO_2$ concentration at each height.