# Peer review of "Nitrogen dioxide stratospheric column at the subtropical NDACC station of Izaña from DOAS, FTIR and satellite instrumentation"

_Atmospheric Measurement Techniques, 2016_

## Referee Comment (RC1) · Anonymous Referee #1 · 30 Mar 2016

Before proceeding with some minor comments, I suggest the authors address these two major, to my mind, issues:

1. Independent studies (Belmonte Rivas et al. 2014; van Geffen et al. 2015; Marchenko et al. 2015) show that, even after the appropriate diurnal correction the SCIAMACHY-nadir and OMI stratospheric VCD(NO2) systematically differ by 8-12*10^14 molec*cm-3 in the contamination-free areas. Such differences do not show any discernible longitudinal dependencies, hence they could be applicable to the Atlantic-ocean area under consideration. However, this study provides the initial (uncorrected for the diurnal NO2 changes) estimates of OMI-SCIAMACHY $\sim 4*10\hat{\ }14$ (bottom sections of Fig. 4), i.e., far below van Geffen's et al. (2015, AMT, 8, 1685) evaluation: OMI-SCIAMACHY $\sim 13*10\hat{\ }14$. The authors should address the source of this $\sim 300\%$ difference in the estimates.

2. I am uncomfortable with the idea of applying the diurnal correction via introduction of a fixed, purely geometric factor. This factor (effective SZA) is based on the 27.5 km estimate of the 'effective height' of the vertical NO2 profile. Firstly, for the tropical zone the SCIAMACHY profiles point to maxima in the NO2 profiles at H>$\sim$30 km (Bauer, R., et al. 2012, AMT 5, 1059). Moreover, the effective height of the stratospheric NO2 shows $\sim$10% seasonal changes (Spinei, E., et al., 2015, AMT, 7, 4299). If the authors insist on introducing the 'effective SZA' correction, then it must be based on the most recent NO2 profile estimates (either models or observations), plus their seasonal changes, since such correction factor should be calculated with the weights provided by seasonal NO2 profiles extracted either from the SCIAMACHY data or the CTM output. In addition, such profile-weighted 'effective SZA' may result in slightly different corrections for the AM and PM observations, based on the differences in the morning/evening NO2 profiles. Besides this rather technical detailing which may introduce some relatively minor adjustments to the proposed correction, I question the validity of the 'effective SZA' approach. At the moment of a twilight zenith observation, the most probable light path (ch. 9, Platt & Stutz, 2008) is defined by two factors: the Rayleigh scattering and the trace-gas absorption, with dominance of the former in the particular NO2 case. True, the strength of absorption depends on the pathway-vise distribution of the absorbers (hence the proposed 'effective SZA'). However, the registered signal is weighted by the intensity of the Rayleigh-scattered light. It remains to be proved (presumably, by applying a single-scattering 2-component RT model) that the 'effective SZA' estimates have some merit. Any RT+CTM -based correction seems to be preferable over the proposed 'effective SZA' factor.

---

## Author Comment (AC1) · 16 Apr 2016

1. Izaña Observatory is located at 2370 m.a.s.l. well above the marine boundary layer. DOAS measurement on horizontal path at the level of the station show concentration of 20-40 pptv (Gómez et al., 2014, Gil-Ojeda et al., 2015). As a consequence stratospheric products have been used for this work, as it is described in the satellite instrumentation section. OMI data are from NASA algorithm (OMNO2 (Collection 3), web page: http://avdc.gsfc.nasa.gov) and SCIA data are from Bremen University (IUPB, v2.0, web page: http://www.iup.uni-bremen.de/doas/scia_no2_data_acve.htm).

[Figure]

Differences in the mean value of those two datasets for the 2005-2011 period are of 18% (OMIstrat=1.18*SCIAstrat). If we take into account that OMI overpass the station few hours later than SCIA, and that represent a mean increase of the column of 12% according to photochemical models, the observed real difference between instruments is of only 6% for the Canary island area. This is what we can say on this issue.

2. AMF used in DOAS technique are, essentially, geometric calculations. Traditionally, corrections to reach the common reference for instruments measuring photochemically changing species at twilight and daytime have been based in NO2 differences between szas calculated from simple boxmodels. The effective AMF is useful since it accounts for the fact that when DOAS GB measure at 90°, the bulk of NO2 is at lower sza. Is a simple correction and the improvement in the comparison between instruments is obvious from figure 4. Concerning the use of a single height for the corrections, we do not claim that this approach is valid for all stations. At the subtropics, the winter-summer temperature excursions is low (4-5K at 10 hPa) and changes in the NO2 height are also small (Spinei et al. 2015 data are from latitudes 46°-52°). The calculations have been carried out assuming the maximum of the layer at 30km, as reviewer correctly points out (see caption figure 2). Line 245 is therefore misunderstanding and we will remove it. The profile selection for the AMFs is always a source of uncertainty. Not only the height of the maximum but the changing NO2 vertical distribution between AM and PM. However as the reviewer mentions, the impact in the AMF has minor effect in the result. We agree to add a sentence such as "at high sza, the slant radiation observed in the surface crosses stratospheric layers of different SZA. Strictly, the correction should be applied separately for each layer. To simplify, a layer representative of the slant path is the stratosphere has been selected."

---

## Referee Comment (RC2) · Anonymous Referee #2 · 28 Apr 2016

This paper presents a 13-year (2000-2012) comparison study of the stratospheric NO2 vertical column density derived from ground-based DOAS and FTIR as well as satellite nadir (OMI, SCIAMACHY) observations over the NDACC station of Izana (Canary island). The differences between the instruments in terms of spatial representativeness and vertical sensitivity are discussed and taken into account in the comparisons. The difference in measurement time, which can have a significant impact in the comparison results in the case of a rapidly photolysing species such as NO2, is corrected by using box-model simulations. The paper shows the importance of considering the effective solar zenith angle of the DOAS observations when comparing them to direct-sun FTIR

and satellite nadir measurements. A first trend analysis performed on all time-series shows an increase of the stratospheric NO2 vertical column but larger trend values are obtained for ground-based instruments than for satellites. Possible reasons for positive trend values and discrepancy between instruments are discussed.

The paper of Robles-Gonzalez et al. is clearly structured and the method and results are generally presented and discussed in an appropriate and balanced way. Therefore I recommend the paper for publication in AMT after addressing the following specific comments:

Specific comments:

Trend analysis: To my opinion, there are several issues with the trend analysis. A first point is that there were instrument changes during the 2000-2012 period for both FTIR and zenith-sky DOAS. Did you investigate the possibility to have a bias in the corresponding NO2 vertical column time-series due to these instrument changes ? If not, this should be done and if there is a bias for one or both techniques, then its impact on the trend analysis should be assessed. A second point is that applying a linear regression for the trend analysis is maybe too simplistic for a time-period of 13 years. I think it would be useful to include the solar cycle and QBO in the calculation. This would also help to compare the derived values with other published studies since the latter take usually these effects into account.

Comparison DOAS/FTIR: it is found that AM values compare better than PM ones. A possible reason for that would be the contamination of afternoon FTIR measurements by the upwelling of high NO2 boundary layer airmasses. Maybe this effect could be quantified for some selected days.

The maximum of NO2 vertical column is observed in June for satellite and in July for ground-based instruments. Did you investigate the impact of the temperature dependence of the NO2 cross sections on these results ? Do you obtain similar results without applying any photochemical correction on the different data sets ?

[Figure]

Technical corrections:

The overall quality of the English is poor throughout the manuscript. Maybe the authors should think to polish the text with the help of an English native colleague.

Some other technical corrections:

*Page 2, lines 70-71: 'GB DOAS' instead of 'DOAS GB' *Page 2, line 74-75: you should add units to the difference values (molec/cm2) *Page 4, line 138-140: 'The air mass factor (AMF) used for the conversion of NO2 slant columns to vertical columns are the NDACC NO2 standard AMF available on the NDACC UV-vis web page (http://ndacc-uvvis-wg.aeronomie.be/tools.php) and based on the Lambert et al. (1999) harmonic climatology of NO2 profiles' *Page 7, line 248: 'the DOAS effective airmass' -> 'the location of the DOAS effective airmass' *Page 8, section7: replace 'diagonal' by '1:1 line'. *Page 17, Table 1: 'BIRA-IASB NO2 AMF LUTs' -> 'NDACC NO2 AMF LUTs' *Page 18, Figure 1: units for NO2 concentration should be molec/cm3 and not molec/cm2.

---

## Referee Comment (RC3) · Anonymous Referee #1 · 10 May 2016

The author's reply does not address my initial concern regarding the effective SZA correction:

- The proposed ESZA correction is meritorious for species with relatively short photochemical lifetimes, $NO_2$ in particular. Considering the sensitivity of the issue (e.g., ∼30% $NO_2$ changes between SZA=90 and SZA=86.8 cases around the winter solstice), there is a lack of detail about the ESZA evaluation. Using some overly simplistic assumptions, one may arrive at EZSA∼85 instead of the used ESZA=86.8. Unless all the underlying details and assumptions are explicitly mentioned by the authors, it is

impossible to judge validity of the approach.

- The same applies to the effective (projected) DOAS pathway used in the DOAS-satellite collocations. Why the authors arrive at the 300 km estimate instead of, e.g., ~360 km (again, using, for a sake of argument, overly simplistic, purely geometric assumptions and ESZA=86.8)? Is there some optical-pathway weighting applied by the authors? [There should be some.] Please provide more details.

Additional remarks:

- Figure 4, bottom-left panel: Why the DOAS-FTIR SZA=90 difference is negative? I see a positive shift in the upper-left panel. Two panels contradict each other. Now, going to Table 2, I also see the negative SZA=90 DOAS-FTIR shift. The negative sign also quoted in the text (Section 7) and re-confirmed in Figure 6. This contradiction must be resolved. What FTIR and DOAS data are used in the plot and the stats (Table 2)? AM? PM? Both?

- Please clarify the AM/PM split in the FTIR data (Fig. 5). Is this related to how the FTIR data are referenced to the either DOAS-AM or DOAS-PM observations? Or you really subdivide the FTIR records into the AM and PM parts? If the latter is true, then Fig. 6 should have two FTIR points. So does Table 3. Please be explicit in description of the data sets in Table 3: e.g., does the OMI-DOAS mean OMI-DOAS(AM), or OMI-DOAS(PM), or something else?

- The caption of Figure 10 mentions two panels, (a) and (b). I see only one.

- Lines 400-405. The more pronounced $NO_2$ trends seen in the DOAS observations are ascribed to the relatively higher DOAS sensitivity to the lower-stratospheric $NO_2$ concentrations. How does this questionable conclusion come along with the factor-of-three lower changes detected by SCIAMACHY and OMI, despite their comparatively high strat-trop sensitivity (cf. the DOAS and satellite AVKs in Fig.1)? It seems, in this respect the DOAS observations are the only outstanding category, since both FTIR

and satellites deliver comparable results, despite their different sensitivity to various stratospheric NO2 layers.

---

## Author Comment (AC2) · 14 Jun 2016

The author's reply does not address my initial concern regarding the effective SZA correction: - The proposed ESZA correction is meritorious for species with relatively short photochemical lifetimes, $NO_2$ in particular. Considering the sensitivity of the issue (e.g., about 30% $NO_2$ changes between SZA=90 and SZA=86.8 cases around

the winter solstice), there is a lack of detail about the ESZA evaluation. Using some overly simplistic assumptions, one may arrive at EZSA of about 85 instead of the used ESZA=86.8. Unless all the underlying details and assumptions are explicitly mentioned by the authors, it is impossible to judge validity of the approach. - The same applies to the effective (projected) DOAS pathway used in the DOAS satellite collocations. Why the authors arrive at the 300 km estimate instead of, e.g., about 360 km (again, using, for a sake of argument, overly simplistic, purely geometric assumptions and ESZA=86.8)? Is there some optical-pathway weighting applied by the authors? [There should be some.] Please provide more details.

The selection of a single point as representative of the layer is a simplification since that "effective point" changes with the sza, the day of the year and the dynamics (through vertical displacement of the layer). We have made use of the Lambert-99 climatology for the latitude based on HALOEv19 and POAM-II data to establish the height of the bulk of the layer, or more precisely, the effective height as the mean height weighted by the $NO_2$ concentration using a annual mean profile. In the Northern Subtropical region the "effective height" does not vary much and so the "effective sza", either. In figure 2c the effect of the height assumption on the effective sza is shown. A change of 5 km in height of the bulk of the layer (25km to 30 km) at sza 90° over the station makes the "effective sza" to change from 86.4° to 87.2°. (We don't see how the reviewer arrives to 85°). The error due to such a change would be $\pm$ 0.4°, that means 1.4-1.5% in the column, depending on the season.

The same applies for the scanned airmass projection. Following our "best geometric estimation" for the $NO_2$ distribution over the station, we reach to approximately 300 km radius. Whether it is 300 or 360 km has a negligible meaning for the purpose of the work, in a Subtropical station. We have, nevertheless computed such a difference providing that it is expected some $NO_2$ column increase around the archipelago due to human activities than in the areas outside their influence.

We will clarify this issue by changing the text in the manuscript ". Line 245. "As the

stratospheric NO2 layer is centered at about 25-30 km height (orange layer in figure 2a),..."

Line 270. "To compute it, the mean height of the NO2 layer weighted by the concentration in a mean NO2 profile has been used. The mean vertical distribution above 17 km was obtained by annual averaging of mean morning profiles from the HALOE and POAM-II data (Lambert 1999) whereas for lower latitudes the output of the SLIMCAT boxmodel was used (Denis et al., 2005). No tropospheric NO2 has been considered. In the Northern Subtropical region the "effective height" does not vary much and so the "effective sza", either. In figure 2c the effect of the height assumption on the effective sza is shown. A change of 5 km in height of the bulk of the layer (25km to 30 km) at sza 90° over the station makes the "effective sza" to change from 86.4° to 87.2°. The error due to such a change would be ± 0.4°, which means 1.4-1.5% in the column, depending on the season. We estimate this error as the upper limit.

Ref: Denis, L., Roscoe, H. K., Chipperfield, M. P., Van Roozendael, M., and Goutail, F.: A new software suite for NO2 vertical profile retrieval from ground-based zenith-sky spectrometers', J. Quant. Spectrosc. Ra., 92, 321–333, 2005

Additional remarks: - Figure 4, bottom-left panel: Why the DOAS-FTIR SZA=90 difference is negative? I see a positive shift in the upper-left panel. Two panels contradict each other. Now, going to Table 2, I also see the negative SZA=90 DOAS-FTIR shift. The negative sign also quoted in the text (Section 7) and reconfirmed in Figure 6. This contradiction must be resolved. What FTIR and DOAS data are used in the plot and the stats (Table 2)? AM? PM? Both?

We thanks the reviewer for noticing the mistake. There was a wrong sign in the plot in both the DOAS and the FTIR data. Only AM data were used here.

- Please clarify the AM/PM split in the FTIR data (Fig. 5). Is this related to how the FTIR data are referenced to the either DOAS-AM or DOAS-PM observations? Or you really subdivide the FTIR records into the AM and PM parts? If the latter is true, then

Fig. 6 should have two FTIR points. So does Table 3. Please be explicit in description of the data sets in Table 3: e.g., does the OMI-DOAS mean OMI-DOAS(AM), or OMIDOAS(PM), or something else?

For the comparison of the DOAS-FTIR data records are subdivided into the AM and PM values while OMI-DOAS and SCIA-DOAS are refereed only to the DOAS AM data. The data values used in Figure 4 for the comparison of DOAS and FTIR are only AM data. This issue will be clarify in the text, in line 290 we will changed the text by this one: "In our work the stratospheric $NO_2$ from FTIR AM data, SCIAMACHY and OMI instruments has been photochemically corrected to the DOAS AM measurement time while FTIR PM data has been corrected to the DOAS PM using the BIRA-IASB (Belgian Institute for Space Aeronomy)..." And in line 306 we will replace the sentence "Figure 4 displays the cross-correlation of FTIR, SCIAMACHY and OMI data" with this one "Figure 4 displays the cross-correlation of FTIR (AM data), SCIAMACHY and OMI data"

- The caption of Figure 10 mentions two panels, (a) and (b). I see only one. The part (b) has been eliminated.

The comments about panel (b) has been eliminated.

- Lines 400-405. The more pronounced $NO_2$ trends seen in the DOAS observations are ascribed to the relatively higher DOAS sensitivity to the lower-stratospheric $NO_2$ concentrations. How does this questionable conclusion come along with the factor-of three lower changes detected by SCIAMACHY and OMI, despite their comparatively high strat-trop sensitivity (cf. the DOAS and satellite AVKs in Fig.1)? It seems, in this respect the DOAS observations are the only outstanding category, since both FTIR and satellites deliver comparable results, despite their different sensitivity to various stratospheric $NO_2$ layers.

We accept the reviewer comment on the explanation provided in the manuscript. At the moment we have no solid explanation for the large differences in trends between DOAS

and FTIR, SCIA and GOME and as a consequence the following paragraph starting in line 605 will be removed from the manuscript: Figure 11 shows the ratio of sensitivities in the stratosphere between DOAS and FTIR, weighted by the concentration at each height. It is found that DOAS sensitivity is higher than FTIR in the lower stratosphere (below 28 km) whereas the opposite is true above that region. Larger trends in DOAS mean that the increase takes place in the layer in which NOx is dynamically controlled, also playing in favor of the above mentioned explanation.

We would like to mention, however, that NO2 trends observed by DOAS are very similar to NOy trends measured by MIPAS, and also calculated by the WCAAM model (+8.5%/decade, and over 20%/decade at 25 km). These trends have been shown in meetings but not yet published in a peer review paper (Funke et al., 2015). The trend section will be object of a future dedicated paper. Ref: Funke, B., Lopez-Puertas, M., Stiller, G., von Clarmann, T. and Gacia, R.: Stratospheric NOy: global budget and vaiability in 2002-2012 from MIPAS observations, in Regional SPARC workshop, Granada, Spain

Funke, B. Stratospheric NOy: Global budget and variability in 2002-2012 from MIPAS observations, 26th IUGG General Assembly 2015, June 22-July 2, 2015, Prague, Czech Republic, Symp. 14 (Middle Atmosphere Science), available at http://www.czech-in.org/cmdownload/IUGG2015/presentations/IUGG-1701.pdf.
* * *

---

## Author Comment (AC3) · 14 Jun 2016

Trend analysis. A first point is that there were instrument changes during the 2000-2012 period for both FTIR and zenith-sky DOAS. Did you investigate the possibility to have a bias in the corresponding $NO_2$ vertical column time-series due to these instrument changes ? If not, this should be done and if there is a bias for one or both techniques, then its impact on the trend analysis should be assessed.

DOAS NDACC dataset has been carefully homogenized in last years through a number of EU framework program projects (i.e. GEOMON, NORS). Data were reprocessed following the NDACC recommendations by using same analysis, same cross-sections and same AMF code. Most important change was the switching from PMT scanning spectrometer to PDA-detector spectrometer in late 1998. However, differences between both instruments during 3-years overlapping period were negligible (slope=0.997, r2=0.96 standard deviation=1.4E14 molec.cm-2) and therefore no correction factors were needed (Gil et al., 2008). The overlapping period is already mentioned in line 118, but will make it more clear in the text, by changing a few things in paragraph starting in line 115 and adding in line 128 the next paragraph: "A 3-year overlapping period was used to ensure the serie continuity. However no corrections to the data were needed since the agreement between instruments was excellent (see Gil et al. 2008). A more detailed description of the instrument can be found in Gil et al., 2008.".

A second point is that applying a linear regression for the trend analysis is maybe too simplistic for a time-period of 13 years. I think it would be useful to include the solar cycle and QBO in the calculation. This would also help to compare the derived values with other published studies since the latter take usually these effects into account.

We agree with the reviewer that the analysis trend is very simplistic. A detailed trend analysis based on multiple regression from a number of stations is on the way for a future publication. We find, however, interesting to compare all satellite and GB datasets available even with such a simple approach since at this particular station the evolution is dominated by the seasonal waves. We have included a sentence to reflect this more clear in the manuscript, in line 396: ". The fact not to take these two effects into account would imply a possible inaccuracy over some stations but over our study station the evolution of the NO2 is dominated by the seasonal waves, therefore, the omission of the QBO and the stratospheric temperature has a minor effect due to the fact that we are considering the most relevant one. Anyway a more detailed study on NO2 trends is ongoing in order to improve the preliminary trends presented here and to better understand the results."

Comparison DOAS/FTIR: it is found that AM values compare better than PM ones. A possible reason for that would be the contamination of afternoon FTIR measurements by the upwelling of high NO2 boundary layer airmasses. Maybe this effect could be quantified for some selected days.

We suggest in the text two possibilities including the pollution upwelling. In recent work (Gil-Ojeda et al ACP, 2015) it was shown that significant upwelling due to slope heating increase the NO2 concentration at the level of the station during the day. The relative importance of this effect on direct sun spectroscopic measurements is dependent on the thickness of the polluted layer above the station but also on the sensitivity of the instrument to lower layers. FTIR sensitivity to lower troposphere is very poor and consequently we will remove the possibility of contamination on FTIR data. Even though some efforts have been done to clarify the reason of the PM discrepancy, at present we have no explanation to provide.

The maximum of NO2 vertical column is observed in June for satellite and in July for ground-based instruments. Did you investigate the impact of the temperature dependence of the NO2 cross sections on these results?.

This is one of the surprising findings of the work. Ground based instruments use NO2 cross section at 220K temperature (Vandaele et al) all year round. This might introduce a bias in high latitudes data. However, temperature seasonal excursions are low in the tropical lower stratosphere. The June to July difference at 10 hPa is of 1K and even smaller at lower heights. The temperature dependence in cross sections is of 2-3%/10K temperature change and therefore it cannot explain the observed discrepancy.

Do you obtain similar results without applying any photochemical correction on the different data sets?

The same seasonal behavior is observed if data are not corrected photochemically. There is a chance that it could be related to the climatology used in the AMF calculations for the location, which uses monthly values, therefore it is different for June and

July. We will explore that possibility. However, FTIR-direct Sun does not use the same climatology, the FTIR uses daily pressure and temperature profiles